# Coupling Quantum Random Walks with Long- and Short-Term Memory for High Pixel Image Encryption Schemes

**DOI:** 10.3390/e25020353

**Published:** 2023-02-14

**Authors:** Junqing Liang, Zhaoyang Song, Zhongwei Sun, Mou Lv, Hongyang Ma

**Affiliations:** 1School of Information and Control Engineering, Qingdao University of Technology, Qingdao 266033, China; 2School of Environmental and Municipal Engineerin, Qingdao University of Technology, Qingdao 266033, China; 3School of Science, Qingdao University of Technology, Qingdao 266033, China

**Keywords:** image encryption, high pixel density, neural networks, quantum random walk

## Abstract

This paper proposes an encryption scheme for high pixel density images. Based on the application of the quantum random walk algorithm, the long short-term memory (LSTM) can effectively solve the problem of low efficiency of the quantum random walk algorithm in generating large-scale pseudorandom matrices, and further improve the statistical properties of the pseudorandom matrices required for encryption. The LSTM is then divided into columns and fed into the LSTM in order for training. Due to the randomness of the input matrix, the LSTM cannot be trained effectively, so the output matrix is predicted to be highly random. The LSTM prediction matrix of the same size as the key matrix is generated based on the pixel density of the image to be encrypted, which can effectively complete the encryption of the image. In the statistical performance test, the proposed encryption scheme achieves an average information entropy of 7.9992, an average number of pixels changed rate (NPCR) of 99.6231%, an average uniform average change intensity (UACI) of 33.6029%, and an average correlation of 0.0032. Finally, various noise simulation tests are also conducted to verify its robustness in real-world applications where common noise and attack interference are encountered.

## 1. Introduction

With the rapid development of Internet technology, more and more high-value data and information is being transmitted over the Internet, and therefore the security of data transmission is becoming more and more important. While ordinary data can be hidden and protected by classical encryption schemes such as DES [1] and AES [2], the information contained in an RGB image is represented by the pixel values. Because of the strong correlation between the neighbouring pixel values of RGB images and the amount of information stored in images, classical encryption schemes are often unable to achieve good encryption of image information, so the encryption of image information is separated from classical data encryption and becomes a separate research direction, focusing on image specific encryption schemes from the data information characteristics of images [3,4,5,6,7,8]. One very promising direction is the application of neural networks to image encryption. This is because cryptography places particular emphasis on the introduction of nonlinear transformations, which is a distinctive feature of neural networks, and, in addition to this, neural networks have characteristics such as ultra-fast parallel processing and operate in matrix form, all of which are extremely well suited to the field of image encryption, making neural networks increasingly interesting in the field of image encryption [9,10,11].

The LSTM [12] is a special type of recurrent neural network (RNN) [13] that uses the ’inner loop’ of a neural network to preserve the contextual information of a time series, allowing the use of past signal data to infer an understanding of the current signal. Theoretically, RNN can retain information from any moment in time. However, in practice, the transfer of information tends to decay over long time intervals, and the effectiveness of the information is greatly reduced after a certain period of time. As a result, RNN is not well equipped to deal with the problem of long-term information dependence, resulting in a tendency to rely only on the most recent input information for inference. To overcome this problem, LSTM is proposed to solve the long-term dependency problem. In contrast to RNN, remembering the content of earlier moments is its default behaviour. Therefore, it does not require a significant cost specifically and works better.

Quantum computing is a new computing mode that follows the laws of quantum mechanics to regulate quantum information units for computing [14]. Quantum algorithm [15,16,17,18] is an algorithm based on quantum computation. By using the unique behavior of quantum mechanics, such as superposition, entanglement, and interference, some algorithms have achieved exponential acceleration compared with classical algorithms [17,19]. Quantum random walk (QW) is a quantum algorithm, which was first proposed by Aharonov et al. [20], including continuous time QW [21] and discrete time QW [22]. Compared with the classical random walk, the algorithm has a significant improvement in computational efficiency, and its time complexity is reduced from O(*n*${}^{2}$) to O(*n*). On the basis of one-dimensional QW, Baryshnikov et al. studied the difference between two-dimensional and one-dimensional coordinate space, and expounded the advantages and unique properties of two-dimensional QW [23]. Although QW is a quantum algorithm, its probability matrix can be solved by classical computers, and the algorithm complexity is still O(n), which makes QW be able to be applied in classical computers in advance.

Both LSTM and QW have applications in image encryption. He et al. [24] proposed an OF-LSTMS that replaces the matrix operation in LSTM with an XOR operation to obtain an encrypted image after a single forward propagation. Yang et al. [25] studied the properties of one-dimensional QW and applied it to quantum image encryption for the first time. Abd et al. [26] analyzed the statistical properties of the probability distribution matrix of two-dimensional quantum walks and applied it to image encryption; Ma et al. [27] combined the discrete cosine transform (DCT) [28] and the probability matrix of alternating quantum walks (AQW) for image encryption, etc.

Although QW probability matrices have been widely used in the field of image encryption, they still have shortcomings and are too inefficient when dealing with high pixel images. The time complexity of the one-dimensional AQW probability matrix is O(*n*), and the computational complexity of the AQW probability matrix is O($n^{2}$), which is still polynomial in time complexity, but the time consumed to generate the QW probability matrix is unacceptable in practical applications to encrypt high pixel value images. At the same time, we also found that the statistical properties required for the encryption of the QW probability matrix are not satisfactory, so when QW is used for encryption, other algorithms are often used to improve the encryption, e.g., Ma used a discrete cosine transform algorithm to perform further dislocation encryption in the DCT domain after applying QW to confuse the pixel values. This does not increase the encryption efficiency too much, but the use of separate algorithms for the scrambling and obfuscation phases nullifies the advantage of having an infinite key matrix for the QW, as it can only participate in one of the scrambling and obfuscation phases, and the two phases are independent of each other.

In order to optimize the statistical properties of the QW probability matrix and its performance on high pixel precision image encryption for better encryption, we propose an image encryption scheme that combines neural networks with quantum algorithms. By combining the QW with the LSTM, the initial matrix is generated using the QW probability matrix, and after training through the LSTM, a suitable prediction matrix is output as the key matrix for encryption according to the required pixel accuracy of the image to be encrypted. We show that this combination can improve the efficiency of the key matrix generation, and at the same time, because the QW probability matrix has strong randomness, the LSTM can not effectively find its pattern to predict, so the generated prediction matrix is also disordered, and has better statistical properties than the QW probability matrix for encryption, which can be better used as a key matrix for encryption. Section 2 of this paper presents the basics related to encryption schemes, including the study and analysis of LSTM and AQW. Section 3 presents specific encryption schemes. Section 4 presents the simulation and theoretical analysis of this paper for detecting the effectiveness of the encryption scheme and lists the comparison of similar schemes to the encryption scheme proposed in this paper. Section 5 concludes the work in this paper and also provides an outlook on the subsequent work. The most critical module of the LSTM is the cell state, which is represented by $C_{t}$, the current state at the current moment, and is generated by the state $C_{t-1}$ at the previous moment together with the signal input $x_{t}$ at the current moment, while $C_{t}$ will continue to be passed to the next moment together with $x_{t+1}$ to generate $C_{t+1}$.

## 2. Related Work and Background Knowledge

### 2.1. Long Short-Term Memory

LSTM is a type of Recurrent Neural Network (RNN) that has been widely used in various applications, such as speech recognition, natural language processing, and time series prediction. Unlike traditional RNNs, LSTMs have an internal memory cell that enables them to maintain information over a longer period of time, making them well-suited for tasks that require modeling sequential data with long-term dependencies.

The core component of an LSTM unit is its memory cell, which is responsible for maintaining information over a long period of time. The memory cell is controlled by three types of gates: the input gate, the forget gate, and the output gate. The input gate controls the flow of new information into the memory cell, the forget gate controls the amount of information retained from the previous time step, and the output gate controls the flow of information out of the memory cell and into the hidden state of the LSTM unit.

The LSTM architecture is derived from the equations that govern the behavior of the gates and the memory cell. At each time step, the input, forget, and output gates are computed using a sigmoid activation function, while the memory cell is updated using a tanh activation function. The equations governing the behavior of the LSTM unit are given by:(1)$$i_{t}=\sigma \left(W_{ix}x_{t}+W_{ih}h_{t-1}+b_{i}\right)$$
(2)$$f_{t}=\sigma \left(W_{fx}x_{t}+W_{fh}h_{t-1}+b_{f}\right)$$
(3)$$o_{t}=\sigma \left(W_{ox}x_{t}+W_{oh}h_{t-1}+b_{o}\right)$$
(4)$$c_{t}=f_{t}⊙c_{t-1}+i_{t}⊙\tanh\left(W_{cx}x_{t}+W_{ch}h_{t-1}+b_{c}\right)$$
(5)$$h_{t}=o_{t}⊙\tanh\left(c_{t}\right)$$
where $x_{t}$ is the input at time step *t*, $h_{t-1}$ is the hidden state at the previous time step, $i_{t}$, $f_{t}$, and $o_{t}$ are the input, forget, and output gates at time step *t*, $c_{t}$ is the memory cell at time step *t*, and σ and tanh are the sigmoid and hyperbolic tangent activation functions, respectively.

The LSTM architecture has proven to be highly effective in various applications, due to its ability to capture long-term dependencies and selectively forget or retain information. The equations presented here provide a foundation for understanding the behavior of LSTMs and for developing new models that incorporate LSTM units.

The chain structure diagram of the LSTM is shown in Figure 1, which illustrates the chain relationship between the three adjacent substructures and the composition of each LSTM substructure.

### 2.2. Quantum Random Walk

This paper is based on the theory of discrete-time QW. The discrete-time QW consists of four main elements: the walker, the coins carried by the walker, the coin toss, and the walk rule.

The Hilbert space $\hat{H}$ of a one-dimensional discrete-time QW tensor consists of the walker position space $H_{w}$ and the coin space $H_{\Gamma }$: $\hat{H}=H_{w}⊗H_{\Gamma }$. In a QW, each step of the walk is determined by a unique coin flip operator Γ: (6)$$\Gamma =\left(\begin{array}{cc} \cos\beta & \sin\beta \\ \sin\beta & -\cos\beta \end{array}\right)$$

After the coin toss is completed, the movement of the walker is specified by the conditional displacement operator $S_{i}$: $S_{i}|\hat{x}\rangle =\left|\hat{x}+{\left(-1\right)}^{\Gamma }\right|,\Gamma \in 0,1$ The $|\hat{x}\rangle \left(\hat{x}\in Z\right)$ in the above equation forms the base vector of the walker’s position space; the two base vectors |0〉,|1〉 form the coin space. We specify: when the coin state is |0〉, the walker is manipulated to move one unit in the forward direction; when the coin state is |1〉, the walker is manipulated to move one unit in the reverse direction.

In the AQW used in this paper, the walker controlled by the coin operator alternates between two arbitrarily chosen vertical directions $\tilde{x}$ and $\tilde{y}$, and the walking operator $\hat{U}$ for the whole QW process can be described as:(7)$$\hat{U}={\hat{S}}_{\bar{y}}\left(I⊗H_{\Gamma }\right){\hat{S}}_{\bar{x}}\left(I⊗H_{\Gamma }\right)$$
where ${\hat{S}}_{\tilde{y}},{\hat{S}}_{\dot{x}}$ are the displacement operators of the walker at each point on the $\tilde{x}$ and $\tilde{y}$ axes:(8)$$\begin{array}{cc} {\hat{S}}_{\tilde{y}}= & \sum_{\tilde{x},\tilde{y}}^{N}(|\tilde{x},\left(\tilde{y}+1\right)\;\mathrm{mod}\;\varpi ,0\rangle \langle \tilde{x},\tilde{y},0|)\\ \; & +\sum_{\tilde{x},\tilde{y}}^{N}(|\tilde{x},\left(\tilde{y}-1\right)\;\mathrm{mod}\;\varpi ,1\rangle \langle \tilde{x},\tilde{y},1|)\\ {\hat{S}}_{\dot{x}}= & \sum_{\tilde{x},y}^{N}(|\left(\tilde{x}+1\right)\;\mathrm{mod}\;\varpi ,\tilde{y},0\rangle \langle \tilde{x},\tilde{y},0|)\\ \; & +\sum_{\tilde{x},\tilde{y}}^{N}(|\left(\tilde{x}-1\right)\;\mathrm{mod}\;\varpi ,\tilde{y},1\rangle \langle \tilde{x},\tilde{y},1|) \end{array}$$
where ϖ indicates the prescribed walking boundary.

Suppose the initial moment: The walker’s location is $\left(0_{\tilde{x}},0_{\tilde{y}}\right)$, and the coin is in the superposition state $H_{\Gamma }=\cos\alpha |0\rangle +\sin\alpha |1\rangle$; then, the initial moment system state is:(9)$$|\psi_{0}\rangle ={|\varphi_{0}\rangle }_{w}⊗(\cos\alpha |0\rangle +{\sin\alpha |1\rangle )}_{\Gamma }$$

The system state after a *T* walk can be expressed as:(10)$$|\psi_{T}\rangle ={\hat{U}}^{T}|\psi_{0}\rangle$$

## 3. Algorithm Description

### 3.1. The Encryption Process

#### 3.1.1. Preparation of Quantum Random Walk Probability Distribution Matrix

The data of the corresponding element in the matrix are the probability P(δ,ϑ,T) of the walker appearing at the coordinates $\left(\delta_{x},\vartheta_{y}\right)$ of the location, as can be deduced from the above:(11)$$P\left(\delta ,\vartheta ,T\right)={\left|\langle \delta ,\vartheta ,0\left|{\hat{U}}^{T}\right|\psi_{0}\rangle \right|}^{2}+{\left|\langle \delta ,\vartheta ,1\left|{\hat{U}}^{T}\right|\psi_{0}\rangle \right|}^{2}\left(\delta_{x},\vartheta_{y}\right)$$

The resulting probability distribution matrix *M* and its four sub-matrices $M_{1},M_{2},M_{3},M_{4}$ after equiproportional partitioning are as follows:(12)$$\begin{array}{c} M=\left(\begin{array}{ccc} P_{11} & \ldots & P_{1n}\\ \vdots & \ddots & \vdots \\ P_{11} & \cdots & P_{nn} \end{array}\right)\\ M_{1}=\left(\begin{array}{ccc} P_{11} & \ldots & P_{1\frac{n}{2}}\\ \vdots & \ddots & \vdots \\ P_{\frac{n}{2}} & \cdots & P_{\frac{n}{2}\frac{1}{2}} \end{array}\right)M_{2}=\left(\begin{array}{ccc} P_{1\frac{n}{2}} & \ldots & P_{1n}\\ \vdots & \ddots & \vdots \\ P_{\frac{n}{2}\frac{n}{2}} & \cdots & P_{\frac{n}{2}n} \end{array}\right)\\ M_{3}=\left(\begin{array}{ccc} P_{\frac{n}{2}1} & \cdots & P_{\frac{n}{2}\frac{n}{2}}\\ \vdots & \ddots & \vdots \\ P_{n1} & \cdots & P_{n\frac{n}{2}} \end{array}\right)M_{4}=\left(\begin{array}{ccc} P_{\frac{n}{2}} & \cdots & P_{\frac{n}{2}\frac{n}{2}}\\ \vdots & \ddots & \vdots \\ P_{\frac{n}{2}n} & \cdots & p_{nn} \end{array}\right) \end{array}$$

We set the walker to be at the center of the Hilbert space $\hat{H}$ tensed by $H_{w}$ and $H_{c}$, so the four submatrices $M_{1},M_{2},M_{3},M_{4}$ are centrosymmetric about the point $P_{\frac{n}{2}}$ in the final generation. To prevent the LSTM from learning the rule such that the statistical performance of the final generated key matrix is degraded, in this paper, only $\hat{M}=M_{1}$ is chosen as the required initial pseudo-random number matrix to participate in the encryption.

#### 3.1.2. Preparing the Encryption Key Matrix

**Step 1:** Ensure the reproducibility of the LSTM across devices. (i) Fix the random seeds of each dependency library so that each function is called with the same initial value and random value each time it is trained by the LSTM. (ii) Presetting the dropout function in the LSTM to 0, i.e., not dropping any nodes of the neural network, to ensure that the network model is fixed each time. (iii) Fixed platforms as well as devices, taking the current mainstream pytroch framework as an example, which still cannot guarantee the accuracy of model reproduction under different CPU and GPU pairings, and also requires CUDA environment variable configuration, etc. in order to further reduce uncertainty.

**Step 2:** Generate the LSTM input vector. Divide $\hat{M}$ by column:(13)$$\left(\begin{array}{ccc} P_{11} & \ldots & P_{1\frac{n}{2}}\\ \vdots & \ddots & \vdots \\ P_{\frac{n}{2}1} & \cdots & P_{\frac{n}{2}\frac{1}{2}} \end{array}\right)\to \left(\varphi_{1},\varphi_{2},\ldots \varphi_{\frac{n}{2}-1},\varphi_{\frac{n}{2}}\right)$$

${\hat{M}}^{\prime }$ is obtained by Min-Max normalization of $\hat{M}$:(14)$$\left(\varphi_{1},\varphi_{2},\ldots \varphi_{\frac{n}{2}-1},\varphi_{\frac{n}{2}}\right)⟶\left(\xi_{1},\xi_{2},\ldots \xi_{j}\ldots \xi_{\frac{n}{2}}\right)$$

$\xi_{i}$ is the vector to be input.

**Step 3:** Generate the key matrix required for encryption. Input the vectors $\xi_{i}$ in matrix ${\hat{M}}^{\prime \prime }$ into the LSTM in order for training, and set the LSTM prediction quantity as $\gamma^{2}$ to obtain the prediction matrix ${\hat{M}}^{\prime \prime \prime }$:(15)$${\hat{M}}^{\prime \prime \prime }=\left(\begin{array}{ccc} \varpi_{11} & \ldots & \varpi_{1\gamma }\\ \vdots & \ddots & \vdots \\ \varpi_{\gamma 1} & \cdots & \varpi_{\gamma \gamma } \end{array}\right)$$

Inverse normalization of ${\hat{M}}^{\prime \prime \prime }$ yields $M_{E}$:(16)$$\left(\begin{array}{ccc} \varpi_{11} & \ldots & \varpi_{1\gamma }\\ \vdots & \ddots & \vdots \\ \varpi_{\gamma 1} & \cdots & \varpi_{\gamma \gamma } \end{array}\right)⟶\left(\begin{array}{ccc} \partial_{11} & \ldots & \partial_{1\gamma }\\ \vdots & \ddots & \vdots \\ \partial_{\gamma 1} & \cdots & \partial_{\gamma \gamma } \end{array}\right)$$

In Figure 2, we show the comparison between the predicted data and the expected values formed from the accurate data after training the QW probability matrix as an LSTM training matrix. Subplot a shows the trend in randomness between predicted and expected values; subplot b shows the distribution between specific predicted and expected values.

#### 3.1.3. Image Encryption

The R,G and B channels in our proposed encryption scheme are performed separately, and our encryption algorithm is described in terms of γ×γ pixels of RGB image *I* corresponding to a grey-scale map in the form of matrix $M_{I}$.

**Step 1:** Hide the pixel information in $M_{I}$ by obfuscating the pixel values. Here, we borrow the heteroskedastic algorithm to implement the obfuscation operation:(17)$$M_{I}^{\prime }=M_{I}⊕M_{E}$$

**Step 2:** Generate matrix $M_{E}^{\prime }=M_{E}$, sort the index value matrix Ω of $M_{E}^{\prime }$ in order to obtain $\Omega^{\prime }$, reorder the $M_{I}^{\prime }$ after the confusion operation according to the corresponding position in $\Omega^{\prime }$, and achieve the dislocation of the image by destroying the relationship between adjacent pixel values to obtain $M_{I}^{\prime \prime }$. The schematic diagram of the dislocation algorithm is shown in Figure 3.

### 3.2. The Decryption Process

#### 3.2.1. Preparing the Decryption Key Matrix

We use the probability distribution of the alternating quantum random walk algorithm at each grid point as the basis for generating the random number matrix required for encryption. The probability distribution matrix generated by the alternating quantum random walk has been shown to possess pseudo-randomness [22], i.e., the random number matrix $M^{\prime }=M$ generated twice, provided that the initial parameters including α,β,ϖ are the same. Since we have removed the uncertainty and randomness from the LSTM, the $M^{\prime }$ is processed once according to the encryption process for M, and finally the prediction matrix generated by the LSTM is processed to obtain $M_{D}=M_{E}$.

#### 3.2.2. Decryption of Encrypted Image

**Step 1:** The encrypted image $M_{I}^{\prime \prime }$ is obtained using the inverse permutation $M_{I}^{\prime }$. This process is the inverse of the permutation operation, and the algorithm is shown in Figure 3:

**Step 2:**$M_{I}{}^{\prime }$ for obfuscation reduction to obtain $M_{I}$.

### 3.3. Encryption and Decryption Algorithm Flow Chart

We show the key steps of our proposed image encryption scheme by means of a flowchart, including the generation of the QW probability density matrix, the process of generating the key matrix by LSTM, and the two key steps (scrambling, confusion) of the image encryption and decryption process using the key matrix, as shown in Figure 4.

## 4. Simulation and Analysis

To verify the resistance of the proposed scheme, three RGB images with a pixel size of 2000 × 2000 were encrypted and decrypted according to the proposed encryption scheme, and various statistical analyses were carried out on the encrypted images and the keys used, including histogram analysis, correlation analysis and information entropy analysis for the encrypted images; sensitivity analysis and key space analysis for the key matrix, etc.

### 4.1. Experimental Parameters and Encryption and Decryption Results

We use $\varpi =240,\alpha =\frac{\pi }{23},\beta =\frac{\pi }{41}$ as the start parameters of the QW to prepare a QW probability matrix of size 100×2000, and set the prediction length of the LSTM to 2000, i.e., to generate a key matrix of the same size as the RGB image to be encrypted. The encryption and decryption results are shown in Figure 5.

### 4.2. The Statistical Analysis

#### 4.2.1. Correlation Analysis

Adjacent pixel correlation $R_{AB}$ is used to measure the degree of correlation of adjacent pixel values. Adjacent pixel values in RGB images often have strong correlations in horizontal, vertical and diagonal directions. Image encryption algorithms will destroy this correlation, and the degree of destruction can reflect the effect of encryption algorithms. The closer $R_{AB}$ is to 0, the better the destruction effect is, and the more difficult it is to obtain image information through the relationship between adjacent pixels [27].
(18)$$R_{AB}=\frac{\mathrm{cov}(A,B)}{\sqrt{D(A)}\sqrt{D(B)}}$$
where cov(A,B) is the covariance of A,B, and $\sqrt{D(A)}$ and $\sqrt{D(B)}$ are the standard deviations of A and B, respectively. In this paper, the horizontal, vertical, and diagonal correlations of the three RGB images of Lena, Lemon, and Sakur are compared before and after encryption. The correlation values for the three RGB images are shown in Table 1, and the specific pixel distribution information is shown in Figure 6 and Figure 7.

#### 4.2.2. Histogram Analysis

The histogram provides a visual representation of the statistical data of the pixel values in an RGB image. The histogram of a normal image usually has a distinct statistical pattern, and to resist statistical attacks [25], the histogram of an encrypted image must be as uniform and smooth as possible. The more such criteria are met, the more uniform the pixel distribution is, the less statistical information the image displays, the less information can be accurately predicted, and the more secure the image encryption scheme is [15]. In this paper, the histograms of the RGB three channels of Lena, Lemon, and Sakura images are analyzed separately, and the specific histograms are shown in Figure 8 and Figure 9.

#### 4.2.3. Information Entropy Analysis

Information entropy *H* was proposed by Shannon, the father of information theory, to describe the uncertainty of the occurrence of each possible event of the information source. The pixel values of RGB images range from 0 to 255, so the information entropy H≤8. The closer the entropy value is to 8, the more information it carries and the more resistant it is to statistical attacks [11]. The formula for this is as follows:(19)$$H\left(m\right)=-\sum_{i=0}^{N-1}P\left(m_{i}\right)\log_{2}P\left(m_{i}\right)$$
where $m_{i}$ is the grey scale value and $P\left(x_{i}\right)$ is the probability of $m_{i}$ occurrence. This paper analyzes the information entropy of the R,G, and *B* channels of the three different RGB images of Lena, Lemon, and Sakura. The relevant data are shown in Table 2.

#### 4.2.4. Key Sensitivity Analysis

An effective key sensitivity means that a slight change in the key information will result in a significant change in the encrypted image. The ideal values of NPCR and UACI are 99.61% and 33.46%, respectively [29]. Higher calculated values of NPCI and UACI of an encryption scheme indicate that the encryption scheme is more resistant to differential attacks:(20)$$\Gamma \left(i,j\right)=f\left(x\right)=\left\{\begin{array}{cc} 1, & \;\mathrm{if}\;C_{1}\left(i,j\right)\neq C_{2}\left(i,j\right)\\ 0, & \;\mathrm{otherwise}\; \end{array}\right.$$
(21)$$\begin{array}{cc} NPCR & =\frac{\sum_{i,j}\Gamma \left(i,j\right)}{J\times R}\times 100\% \end{array}$$
(22)$$\begin{array}{cc} UACI & =\frac{1}{J\times R}\left[\sum_{i,j}\frac{C_{1}\left(i,j\right)-C_{2}\left(i,j\right)}{255}\right]\times 100\% \end{array}$$
where ,ℑ,R are the length and width of the encrypted image, Γ(i,j) is the above equation, and C1, C2 are the images after encryption with different keys.

In this paper, the key sensitivity of the R,G and *B* channels of the RGB images of Lena, Lemon, and Sakura were analyzed separately, and the relevant data are shown in Table 3.

#### 4.2.5. The Key Space

The key space refers to the set of all possible keys used to generate the key and determines whether the encryption scheme can resist a brute-force attack. Cryptosystems with a key space size of $2^{128}$ are effective in resisting brute force attacks. The key space calculation for the scheme proposed in this paper is based on quantum effects. Since in quantum theory the position of a particle in a defined space is not deterministic, each position has its probability of existence, only with different probabilities, and this probability can be changed by specifying the size of the space for a QW and the initial walking direction and forward direction. As the walk direction takes values from 0 to 2π and the QW is extremely sensitive to accuracy, the change in probability is infinite as the accuracy of the computer increases, i.e., the key space established based on the QW is infinite.

#### 4.2.6. Explicit Attack

Known plaintext attack: The attacker can recover the key by obtaining the decrypted image and comparing it with the ciphertext image. Since the algorithm in this paper has a good diffusion effect, the difficulty of obtaining the key by this method is close to that of a direct brute force attack, so the encryption scheme in this paper can effectively resist known plaintext attacks.Selective plaintext attack: Assuming that the attacker has gained access to the encrypted machine, he can select an arbitrary number of plaintexts for the encryption algorithm under attack to encrypt and obtain the corresponding ciphertexts. The attacker’s goal is to gain some information about the encryption algorithm through this process that will allow the attacker to more effectively crack messages encrypted by the same encryption algorithm (and associated key) in the future. In the worst case, the attacker can simply obtain the key used for decryption. This scheme is commonly used against public key encryption schemes. The keys in this scheme are not universal, i.e., they are changed periodically, even differently each time, making it impossible for an attacker to obtain valid information.

#### 4.2.7. Time Complexity Analysis

The time complexity analysis of an encryption scheme is an important indicator to evaluate the excellence of an encryption scheme, which will directly affect the encryption efficiency. The time consumption of our proposed scheme consists of two parts, one is the time required to generate the key matrix, and the other is the completion of the image encryption by the key matrix. Although the efficiency of generating the pseudo-random number matrix is important, it is not part of the time complexity of the encryption scheme as it is decoupled from the image encryption process. The encryption time complexity of our proposed scheme consists of a combination of pixel obfuscation and scrambling. The time complexity of this process is O($n^{2}+n$log *n*), as the time consumed by matrix permutation is O($n^{2}$). In summary, the encryption time complexity of our proposed scheme is O($2n^{2}+n$log *n*).

#### 4.2.8. Noise Robustness Testing

During the transmission of image information over the network, information may be lost or misplaced due to packet loss, malicious attacks, and so on. We simulate the continuous loss of image information due to network fluctuations using Gaussian noise and pretzel noise. A malicious attack was simulated using partial block replacement of the encrypted image. Figure 10 shows the decrypted image of the Lena encrypted image with the addition of Gaussian noise, pretzel noise and a clipping attack.

### 4.3. Comparison of Encryption Schemes

In this section, we analyze and compare the use of QW alone, the encryption scheme proposed in this paper, and similar work in recent years in terms of the important measures of average relevance, information entropy, average NPCR, average UACI, and key space size to resist brute-force cracking, the data of which are presented in Table 4.

## 5. Conclusions

We propose a more efficient encryption scheme for the current lack of encryption schemes for high pixel images in the field of image encryption. The probability density matrix generated by the quantum random walk is trained by exploiting the memory learning capability of the LSTM and the nonlinear nature of the quantum random walk. It can take advantage of the nearly infinite key space brought by the quantum random walk algorithm, and also solve the shortcomings of the low generation efficiency of the quantum random walk itself. At the same time, both the permutation and obfuscation processes of our scheme make use of the key space of the quantum random walk, avoiding the shortage of key space in a particular process.

## Figures and Tables

**Figure 1 entropy-25-00353-f001:**
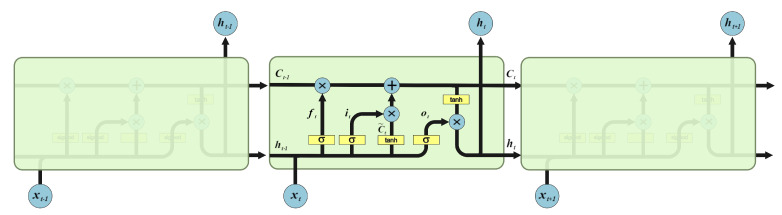
Chain model for LSTM.

**Figure 2 entropy-25-00353-f002:**
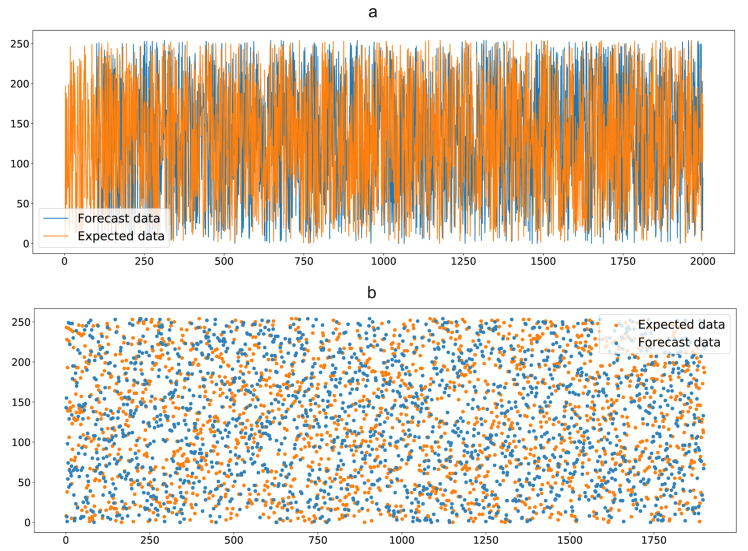
LSTM generation key matrix.

**Figure 3 entropy-25-00353-f003:**
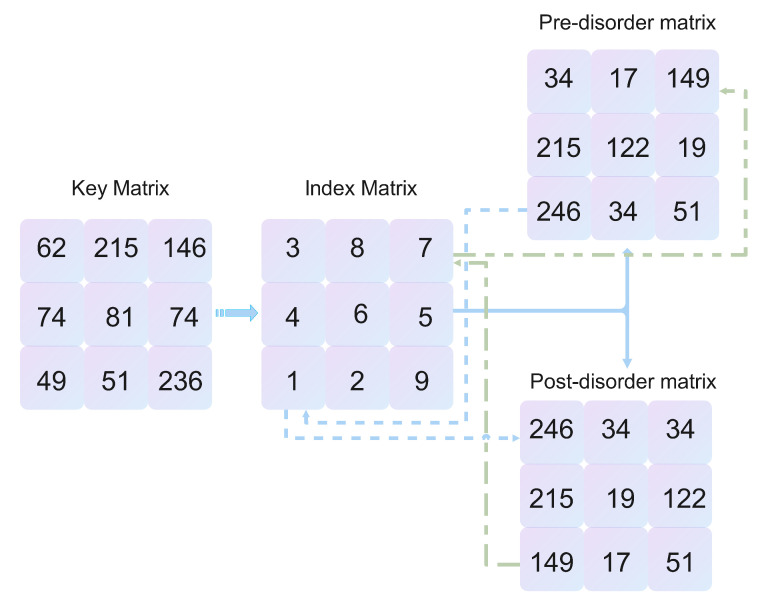
Encryption scheme—scrambling algorithm.

**Figure 4 entropy-25-00353-f004:**
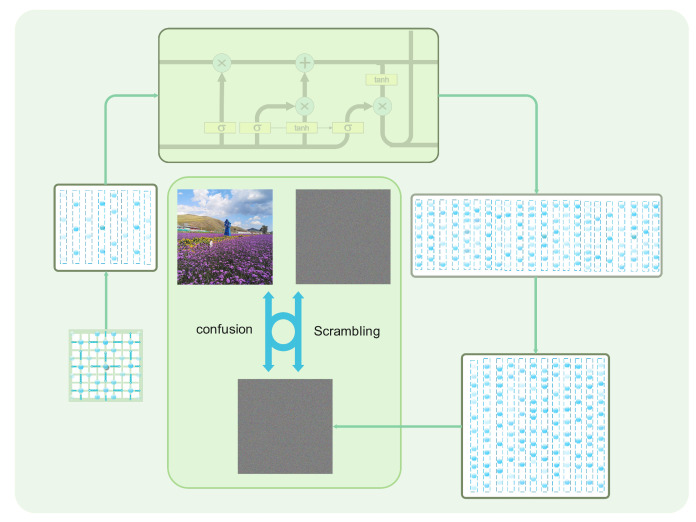
Encryption and decryption process.

**Figure 5 entropy-25-00353-f005:**
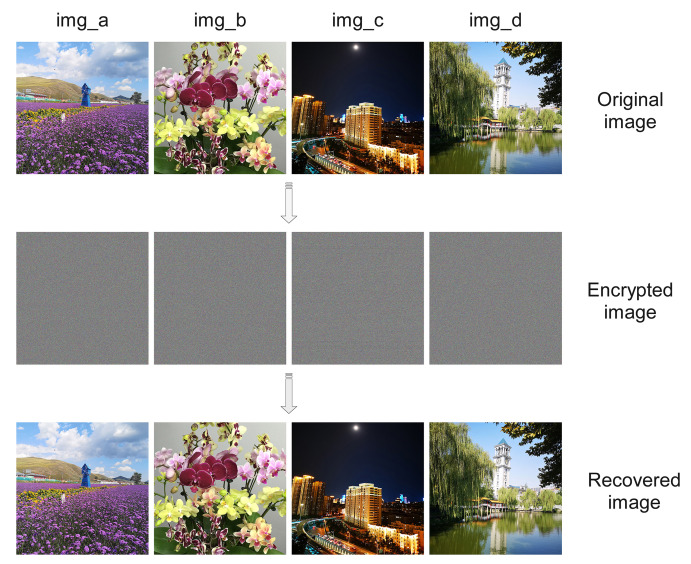
Image encryption before and after comparison.

**Figure 6 entropy-25-00353-f006:**
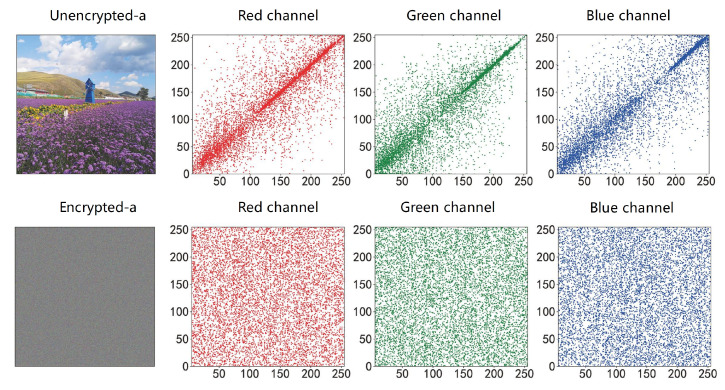
Comparison of correlation before and after img_a encryption.

**Figure 7 entropy-25-00353-f007:**
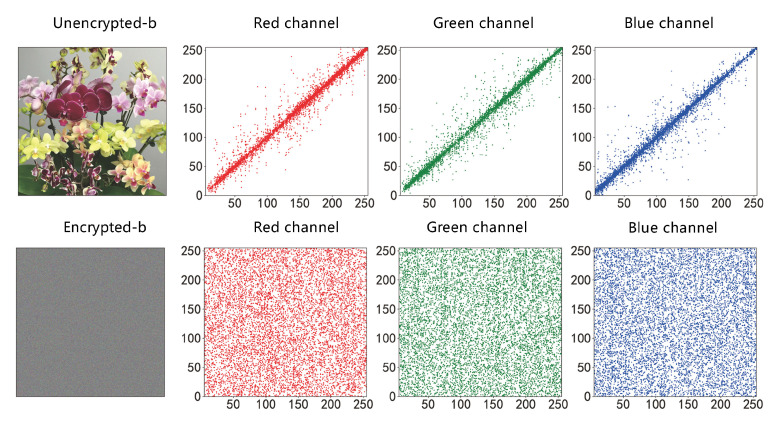
Comparison of correlation before and after img_b encryption.

**Figure 8 entropy-25-00353-f008:**
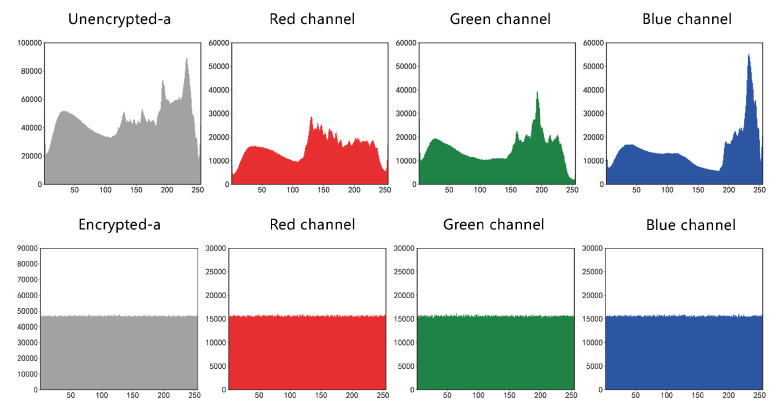
Comparison of histogram before and after img_a encryption.

**Figure 9 entropy-25-00353-f009:**
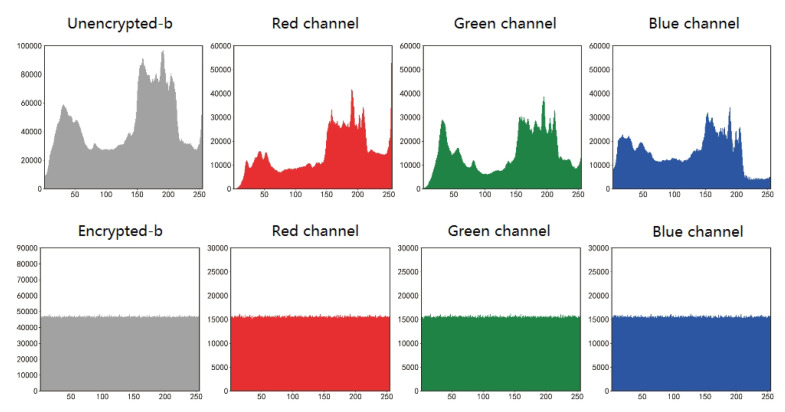
Comparison of histogram before and after img_b encryption.

**Figure 10 entropy-25-00353-f010:**
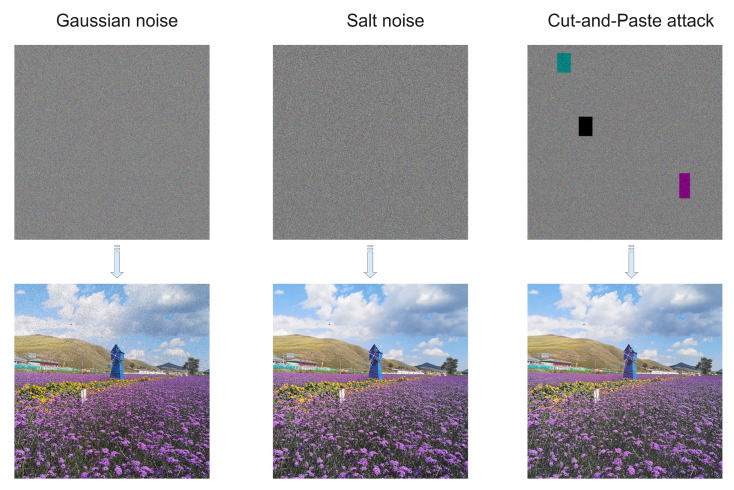
Comparison of histogram before and after img_b encryption.

**Table 1 entropy-25-00353-t001:** Pixel correlation analysis data.

Image	Channel	Horizontal	Vertical	Diagonal
	Red	0.8846	0.8924	0.8297
Unencrypted (img_a)	Green	0.9062	0.9146	0.8568
	Blue	0.9269	0.9272	0.8905
	Red	0.0006	0.0011	0.0032
Encrypted (img_a)	Green	0.0032	0.0027	0.0021
	Blue	0.0041	0.0016	0.0022
	Red	0.9930	0.9944	0.9869
Unencrypted (img_b)	Green	0.9940	0.9949	0.9897
	Blue	0.9927	0.9939	0.9876
	Red	0.0022	0.0011	0.0023
Encrypted (img_b)	Green	0.0021	0.0025	0.0014
	Blue	0.0009	0.0041	0.0013

**Table 2 entropy-25-00353-t002:** Entropy analysis.

Image	Channel	Image Entropy (Bit)
	Red	7.9991
Encrypted (img_a)	Green	7.9996
	Blue	7.9989
	Red	7.9992
Encrypted (img_b)	Green	7.9992
	Blue	7.9994

**Table 3 entropy-25-00353-t003:** Key sensitivity analysis.

Image	Channel	NPCR	UACI
ine	Red	99.6124%	33.4216%
img_a	Green	99.6088%	33.3657%
	Blue	99.6003%	34.2157%
ine	Red	99.6419%	33.6114%
img_b	Green	99.5986%	33.4268%
	Blue	99.6036%	33.5762%

**Table 4 entropy-25-00353-t004:** The comparison in this article is for reference only as the images used in the different solutions are different and have different pixels. As the pixel sizes vary in each scenario, we have used the largest pixel images from their scenarios for comparison and selected their average values as a reference.

Scheme	NPCR (%)	UACI (%)	Correlation	Entropy (Bit)	KeySpace
QW	93.14	32.36	0.0149	7.9947	>$2^{128}$
our	99.6109	33.6024	0.0032	7.9992	>$2^{128}$
[3]	99.6127	33.4471	0.0013		>$2^{128}$
[4]	99.6336	33.4636	0.0026	7.9937	>$2^{128}$
[5]	99.6326	33.4022	0041	7.9973	>$2^{128}$

## Data Availability

The data and information supporting this study can be provided at the request of the corresponding author at reasonable request.

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
