# Peer review of "Coupling Quantum Random Walks with Long- and Short-Term Memory for High Pixel Image Encryption Schemes"

_entropy, 2023, doi:10.3390/e25020353_

Round 1
Reviewer 1 Report
This paper can be accepted in its present form
Author Response
Thank you for your recognition of our work. We take your language questions very seriously and we will ask our professional editors to help with the touch-ups.
Reviewer 2 Report
The paper is interesting (an idea and verification), however some issues should be addressed:
- proofreading by the authors is recommended to avoid some mistakes/generalizations, e.g. “increased from O(n2) to O(n)”;
- also some typos should be corrected and errors in edition (e.g. missing spaces between words and cites);
- some problems with structure of the document – an example is Section 2: Subsubsection 2.1.1. "subsubsection Cell state" is not needed - move it into Subsection 2.1;
- the paragraphs with a single sentence (e.g. in Subsection 2.2) do not look professional;
- I recommend to rewrite ‘background knowledge’ part to allow readers understand the basics (now, it is too complicated);
- explain better the algorithm (Section 3) because some steps are not clear and it is not clear why encryption is truly reversible.
Author Response
Question 1:Ansproofreading by the authors is recommended to avoid some mistakes/generalizations, e.g. “increased from O(n2) to O(n)”.
Response: Thank you for taking the time to read it carefully. We have corrected the error in the latest version and have reorganised the entire text to ensure that the problem no longer exists.
Question 2: Also some typos should be corrected and errors in edition (e.g. missing spaces between words and cites).
Response: We have amended such issues in the new version of the manuscript.
Question 3: Some problems with structure of the document – an example is Section 2: Subsubsection 2.1.1. "subsubsection Cell state" is not needed - move it into Subsection 2.1.
Response: We have fixed the problem in the latest version of the manuscript and have reorganised the paragraphs throughout.
Question 4: The paragraphs with a single sentence (e.g. in Subsection 2.2) do not look professional.
Response: Thank you for pointing this out and we have made structural changes in response to your comments.
Question 5: I recommend to rewrite ‘background knowledge’ part to allow readers understand the basics (now, it is too complicated).
Response: We have rewritten the background knowledge section and presented it as simply as possible to make it easier for the reader to understand.
Question 5: Explain better the algorithm (Section 3) because some steps are not clear and it is not clear why encryption is truly reversible.
Response: We describe our algorithm in further detail, focusing on your emphasis on explaining reversibility, and adding appropriate references to facilitate a more detailed understanding.

Reviewer 3 Report
In the text, the names of the figures are mentioned, whereas in the table, they are indicated by symbols, making it difficult to understand the relationships. The names of the figures should be shown in the tables as well to unify the expressions.
Author Response
Point: In the text, the names of the figures are mentioned, whereas in the table, they are indicated by symbols, making it difficult to understand the relationships. The names of the figures should be shown in the tables as well to unify the expressions.
Response: Thank you for reading it carefully and for your valuable comments. We have standardised the presentation, which really helps to avoid confusion for readers due to naming issues.
